# Moisture Absorption Characteristics and Subsequent Mechanical Property Loss of Enset–PLA Composites

**Abebayehu Abdela** [1,2,3,*] **, Maxim Vandaele** [2] **, Sam Haenen** [2] **, Bart Buffel** [2] **, Belete Sirahbizu** [3] **and Frederik Desplentere** [2]

1   Manufacturing Engineering Chair, School of Mechanical and Industrial Engineering, Ethiopian Institute of Technology-Mekelle, Mekelle University, Mekelle P.O. Box 231, Ethiopia
2   Department of Materials Engineering, ProPoLiS Research Group, KU Leuven Bruges Campus, 8200 Bruges, Belgium; bart.buffel@kuleuven.be (B.B.); frederik.desplentere@kuleuven.be (F.D.)
3   Department of Mechanical Engineering, College of Engineering, Addis Ababa Science and Technology University, Addis Ababa P.O. Box 16417, Ethiopia; belete.sirhabizu@aastu.edu.et
*   Correspondence: abexmesc@yahoo.com

**Abstract:** One of the drawbacks of natural fibers and their composites is their inherent hydrophilic nature. The effect of moisture on the mechanical properties of composites is irrefutable. This study deals with the hygroscopic characteristics of enset–PLA composites and their effect on the mechanical properties of the composites. To do this, injection-molded composite specimens with different fiber volume fractions, plasticizer ratios, fiber lengths, and fiber ages were considered. The specimens were exposed to distilled water, and the moisture absorption was monitored on a daily basis. Subsequently, the specimens were subjected to mechanical loading to determine the effect of moisture on their strength, stiffness, and strain at break strength. Lastly, the individual and joint effects of the considered factors were scrutinized using an optimal experimental design. The results of the study show that the maximum and minimum moisture uptakes were recorded for 25% and 15% fiber ratios, respectively. Due to the effect of moisture, the tensile and bending strength decreased by 11% and 5%, respectively, for the 15% fiber volume fraction and decreased by 16% and 13%, respectively, for the 25% fiber volume fraction. Increasing the amount of plasticizer increases the moisture resistance. The results indicate that Enset–PLA composites have competitive properties and stability when exposed to moisture.

**Keywords:** hydrophilic; PLA; plasticize; fiber concentration

## 1. Introduction

A large number of natural fibers with attractive properties for diverse composite applications are available [1,2]. They are being widely used due to their structural properties and good mechanical characteristics in addition to other benefits, including their natural availability, sustainability, eco-friendliness, and cost advantages [3–5]. Despite these advantages, natural fibers and their composites possess some challenges. A lack of constancy and the variation in their properties as well as their sensitivity to external environments, such as humidity and moisture, are among vital challenges [4–9]. The hydrophilicity of natural-fiber-reinforced composites is one of the major shortcomings that should be examined so as to evaluate their suitability for certain practical applications within different environments [10,11]. In this aspect, the moisture sensitivity of bio-composites emanates from the characteristics of the fiber and the matrix. The presence and ratio of the plasticizer can also be another determinant of the moisture sensitivity of bio-composites [12].

First, natural fibers determine many of the characteristics of bio-composites exposed to moist environments because they are relatively more hydrophilic than polymer matrices [13]. Since fibers are the principal load-carrying and transferring component in fiber-reinforced composites, their mechanical properties play a key role in determining

the resulting characteristics [14–17]. Their properties can be affected by a number of factors. One of the factors that can affect the mechanical properties of the bio-composite is the moisture absorption characteristics of the fiber. Without specific conditioning or drying, the fiber's moisture content usually ranges from 5 to 13% [1,18]. This characteristic of natural fiber plays a critical role, since strongly polarized fibers are inherently hygroscopic. They exhibit poor resistance to moisture, thus leading to high water absorption as well as poor mechanical properties and dimensional stability [19]. In such conditions, the fiber makes polymer impregnation more difficult, causing weak adhesion on the polymer matrix–fiber interface, which leads to internal tensions, porosity, and the premature failure of the system when they are used in a composite [4,10].

Second, biodegradable polymers have relatively poor barrier properties. This characteristic of biodegradable polymers, either in their pure state or when used with natural fibers, requires thorough care [12,20,21]. Most parts of biodegradable polymer systems are made of polyesters; it is well known that polyesters absorb humidity, and the presence of moisture gives rise to noticeable degradation phenomena [11]. This study considered poly lactic acid (PLA) compounded with Enset Fiber (see Section 2.1). Poly lactic acid possesses hydrophilicity and requires drying before processing [10,22]. For instance, the hydrolytic degradation of PLA and a PLA/polycarbonate blend exposed to a high temperature and humidity results in significant moisture absorption and hydrolysis, leading to the degradation of its properties [10,11]. Specifically, the PLA used in this study is PLA 4043D since it absorbs moisture and its data sheet recommends drying before processing. According to this datasheet, a moisture content of less than 0.025% (250 ppm) is recommended to prevent viscosity degradation [23].

Multiple studies attest to the dependence of the mechanical properties of both fibers and their composites on their moisture absorption characteristics [18,19,24]. When exposed to moisture, bio-composites display lower values for their mechanical properties than those of synthetic-fiber-reinforced composites. This happens because moisture affects the performance and physical and mechanical integrity of bio-composites more negatively than synthetic-fiber-reinforced composites [4,21,25]. Consequently, the swelling and shrinkage of the fibers surrounded by the matrix generate internal stresses at the interface and can eventually lead to the matrix significantly degrading the initial properties of the composite [21]. For instance, a decreasing tendency of the tensile strength and the Young's modulus with an increasing relative humidity of flax and nettle fibers has been observed, while a decrease in the Young's modulus of flax fibers by about 23% has been observed when the relative humidity varies from 30 to 80% [26,27].

The rate of the water absorption of bio-composites is another vital feature for comparing materials [28,29]. It differs based on the fiber and the polymer type. The same holds true for PLA-based natural-fiber-reinforced composites. For instance, untreated and acetylated kenaf–PLA composites have shown rapid water absorption on the first day, but the water uptake eventually reached a plateau state. Since the water absorption rate of PLA is less than 1%, the overall result can be interpreted as a result of the water uptake of the kenaf or the interface [5,30–33]. Again, composites of jute, sisal, and elephant grass with PLA display varying moisture uptake levels and different rates of absorption emanated mainly from the fibers' type and their respective hydrophilicity [13]. Additionally, the moisture absorption and resulting effect on the mechanical properties of PLA/natural fiber composites depend on the fiber orientation, fiber volume fraction, the nature of the matrix, and the adhesion between the fiber and the matrix as well as the amount of plasticizer [4,13,34,35].

Thus, the response of bio-composites to different environments depends on the hydrophilic nature of the fiber and the matrix, with the fiber being the major contributor. Similarly, the addition of a plasticizer and its ratio affect the moisture absorption characteristics, resulting in changes in their properties. The effect of the fibers has been consistently reported throughout the literature to be vital to the overall moisture absorption of composites of biopolymers and specifically PLA. Moreover, various fibers have different levels of influence on the hydrophilicity of composites owing to their natures and their varying

properties. Consequently, inherent mechanical properties losses due to moisture absorption are consistently reported by various authors [11,13,14,35]. Therefore, this study considers a PLA–enset composite and defines its hygroscopic characteristics together with the resulting mechanical properties loss.

## 2. Materials and Methods

### 2.1. Materials

Enset fibers (Figure 1) were sourced from Ethiopian indigenous enset plant (*Ensete ventricosum*) from Kokosa, Oromia, Ethiopia, found at an altitude of 2627 m with min and max annual average temperatures of 12 and 18 °C, respectively [36]. Manually extracted fibers from enset plants with 3 different ages (1, 2, and 3 years after the pulp is ready for first round of extraction, approximately 5, 6, and 7 years) using a technique developed in house [1].

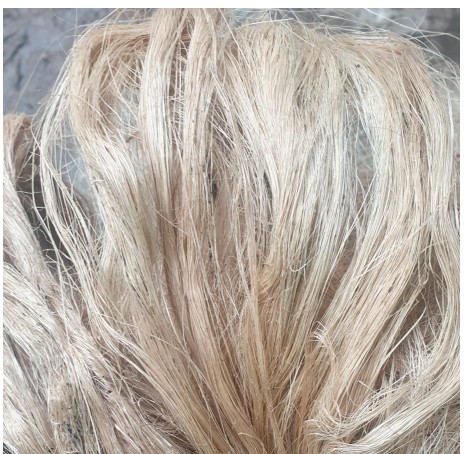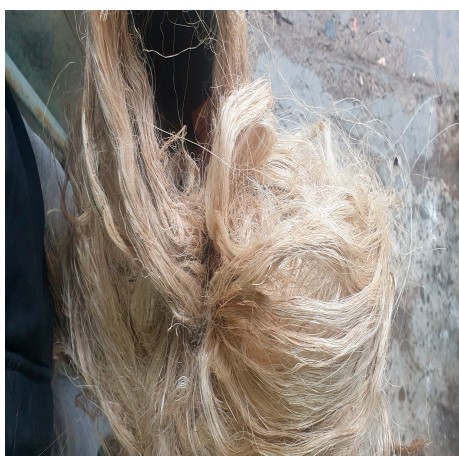

**Figure 1.** Enset Fiber after extraction.

The density of the fibers was determined using a pycnometer (Beckman model 930), for which helium gas at a pressure of 0.5 bar was used as the displacement medium. During material preparation stage prior to the density measurement, the fibers were cut to different sizes based on the requirement mentioned in the methods section below.

Poly lactic acid (PLA 4043D, Nature works) in the pellet form, with a density of 1.24 g/cm$^3$ and melt flow rate of 65 g/10 min, is used for this study. The data sheet shows that the tensile strength, tensile elongation, and tensile modulus are 60 MPa, 6%, and 3.6 GPa, respectively. The flexural strength and modulus from datasheet are 83 Mpa and 3.8 Gpa, respectively.

The plasticizer used to enhance the processability and improve the brittle nature of pure PLA is Proviplast 2624. Different percentages (2, 4, 6, 8, and 10%) of plasticizer were added to the virgin PLA compound based on its effect on the ease of compounding, and the resulting effects on mechanical properties were evaluated to select the optimum percentage in the enset–PLA compound.

### 2.2. Methods

2.2.1. Fiber Density and Moisture Content Estimation

Density of the fiber is measured at different levels and with different conditions based on their requirements. Fibers from milled powder, fibers with 10 mm and 100 mm were vacuum-dried for 6 h and 24 h at 60 °C, respectively. Weight of these fibers is measured using sensitive balance with accuracy of $10^{-5}$ g. Moreover, density of the fiber is measured using helium gas pycnometery. Following density measurement, single fiber characterization is performed and fiber's hygroscopic nature is examined. Afterwards, manually chopped fibers to the desired lengths of approximately 5 mm and 10 mm are prepared and

dried for 12 h in a pressurized air dryer (Moretto) at 60 °C before compounding of the matrix-reinforced plasticizer.

2.2.2. Compounding and Actual Fiber Ratio Substantiation

Compounding is performed for different fiber–matrix–plasticizer combinations based on experiment design. The factors considered are fiber ratio (10, 15, 20, and 25%), plasticizer ratio (2, 4, 6, 8, and 10), fiber plant age (1, 2, and 3), and fiber size (5 mm and 10 mm). The rationale behind considering the fiber ratio and size as factors is derived from other related findings [5,13,26,37–39]. As per these findings, composite strength increases with increasing fiber ratio while loss in properties increases due to increases in moisture and fiber ratio concurrently. Still, the level of effect varies with the fiber type. Again, the need to consider the effect of plasticizer on enset–PLA has been documented for other bio-composites [12,34,40] as well as other factors [1,16,41]. Selection of these factors also must take into account real-world scenarios, at every stage of composite processing as well as manufacturing of the desired bio-composite-based product.

Number of the required experiments became quite large, when combinations of all the above factors were considered. Therefore, number of experiments was reduced when conducting preliminary test to see the level of effects of the aforementioned factors. For instance, the effect of age difference on strength was checked at fiber level before compounding. The result was compared with that of a previous study conducted using digital image correlation technique [1]. No significant strength difference was noted between the fibers, though the 3rd age group showed slightly greater value. Hence, 3rd age group was directly considered.

On the other hand, the fiber lengths were also considered, and the results after compounding and injection molding were checked. Again, slight difference was noted; the one with a length of 5 mm showed slightly greater value. Still, a fiber size reduction was noted during compounding; this was concurrently noted while dissociation of enset–PLA compounds was conducted to find the actual fiber ratio, as shown in Table 1 below. Therefore, the main parameters considered by altering the ratio were the 3rd age group fiber, PLA, and plasticizer. These combinations were manipulated by using the possibility of inputting a given value into the compounding machine. Process of compounding was performed with extruder speed of 160 rpm, melt temperature of 19 °C, and die temperature of 175 °C.

**Table 1.** Comparison of actual and machine-set fiber ratio in the compound.

| Machine Set Ratio (%) | Sample Weight (g) | Filter Paper (g) | Weight after Filtration (g) | Net Fiber Weight (g) | Net Actual (g) | Difference (%) |
|---|---|---|---|---|---|---|
| 15 | 0.373 | 0.455 | 0.501 | 0.046 | 12 | 17.7 |
| 20 | 0.321 | 0.457 | 0.512 | 0.055 | 17 | 14.3 |
| 25 | 0.355 | 0.445 | 0.521 | 0.076 | 21 | 14.4 |

Proportions of enset fiber, PLA, and plasticizer were controlled and altered using the settings on compounding machine. However, finding the actual fiber concentration was also important to compare materials and find the direct impact of the actual fiber ratio on composite characteristics. Consequently, the compound was dissolved in chloroform. PLA dissolved fully in chloroform. Therefore, the resulting value found through dissolving enset–PLA composite was compared with the machine reading, as shown in Table 1 below. While doing this, 140 mL of chloroform was used for a compound that weighs between 8 g and 10 g and dissolved. Full dissolution of the compound took 48 h. Since chloroform is volatile, the drying of the fiber that remained on the filtering paper was fast; the weight was measured after 1 h and 48 h to see if there was residual chloroform. After separation was carried out, net weight of the filtered fiber was calculated against compound weight, and comparison was made between the actual and measured fiber concentration for authenticity.

As is evident from the above table, the actual value of the fiber ratio in the compound is slightly smaller than the machine reading. Its variation falls in between 14.3% and 17.7%, with the variation in the smaller fiber ratio being greater. Then, specimen production is performed according to Section 2.2.3 below.

### 2.2.3. Specimen Production and Testing

Specimens were produced using the injection molding process. Before processing, each compound with different combination of enset–PLA–plasticizer, as shown in Section 2.2.2 above, was dried in pressurized air dryer (Moretto) for 12 h at 60 °C. The drying temperature was kept at 60 °C to avoid sticking of the compounded pellets for higher plasticizer ratio and its effect on subsequent injection molding process. The injection molding was performed at a temperature of 190 °C, a pressure of 1400 bar, and cycle time of 61 s.

Next, the standard tensile bars (dog bone) with known dimensions, including thickness and weight, are dried and submerged in distilled water. Increases in weight are measured daily to check moisture uptake and the absorption trend thereof. Daily monitoring is important to check the absorption trend. Moreover, it allows us to scrutinize the rate of absorption jointly with possibility of finding the asymptotic absorption stability point. Both the rate of absorption and point of stability vary based on the variation in the fiber ratio [13,30]. Saturation point at which there are no more uptakes is identified in this process. Then, the specimens are submerged in distilled water for ten more days and mass are measured to enhance certainty of the results.

After defining hygroscopic behavior of enset–PLA composites, tensile and three-point bending (3 pb) strength of the material was tested to investigate the moisture's effect on strength. The three-point bending test was conducted considering a span length of 80 mm and a rate of 0.01 mm/sec. The result was compared with the strength before uptake to identify associated property losses.

## 3. Results

The moisture content of the fiber extracted at different ages of the enset plant and with different sizes was measured using Karlfisher and Mettler Toledo moisture analyzers. Using the Karlfisher titration method, the measured moisture content of the fiber was 6.34–6.62% (Karlfisher). The moisture content was again measured using a Mettler Toledo moisture analyzer and found to be 5.5–8.2%. In this regard, the fiber extracted from the plants of the matured (third group) age group absorbed slightly less moisture compared with that extracted from the plants in the younger age groups. The moisture contents were 5.5% to 6.4% and 6.36 to 6.4% for the Mettler Toledo and Karlfisher results, respectively.

In addition to their moisture resistance, plants of the third age group had a better level of fiber strength. Consequently, the fiber in this age group was subjected to compounding, injection molding, and submerging in distilled water to finally characterize its subsequent mechanical property losses due to moisture. Conversely, the fiber with a shorter length exhibited a slightly higher level of moisture absorption compared with that of the older age group. This might have resulted from the cutting of the lumen into different parts, exposing a relatively larger surface area and causing porosities to absorb more moisture [15]. This test gave us our first preliminary insight into the better moisture resistance of the enset plant fiber in the older age group. Based on this result, the experiment continued using the fiber from the older age group (because of its slightly better strength and slightly lower moisture absorption) and compounded it with different PLA–plasticizer combinations. The moisture absorption and release characteristics are presented below.

### 3.1. Moisture Absorption and Release Characteristics

The moisture absorption and release characteristics of the enset–PLA composite for different scenarios were studied for 40 and 28 consecutive days, respectively. In this regard, 10 extra days after achieving a relatively stable trend both in terms of absorption and release

were considered to enhance the certainty of our results. The trends are plotted in Figures 2 and 3 below.

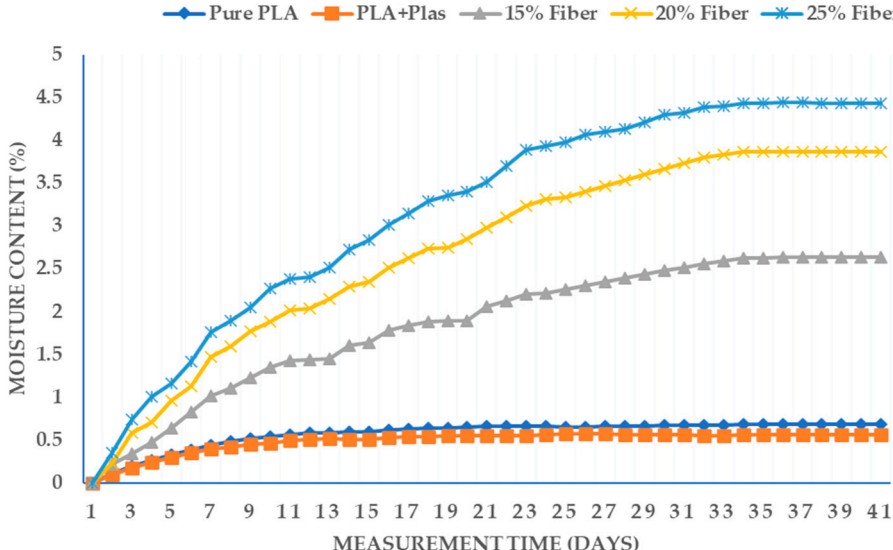

**Figure 2.** Moisture absorption characteristics of enset–PLA composite with 6.25% plasticizer.

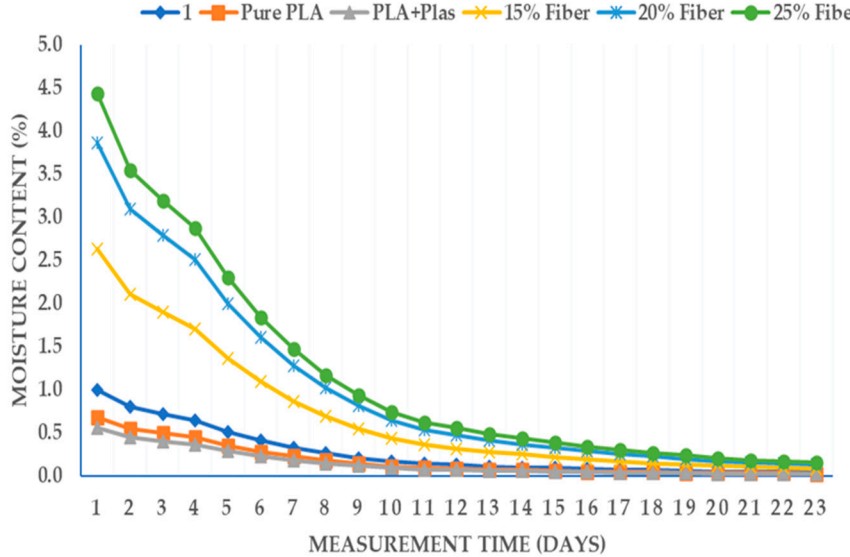

**Figure 3.** Moisture release characteristics of enset–PLA composite with 6.25% plasticizer.

Figure 2 below shows that the moisture absorption is nearly stable after 9 days for the pure PLA specimens and the PLA with a plasticizer. No significant moisture uptake is noted after the 10th day, but 30 more days are considered for the comparison with the PLA composite incorporating the enset fiber. On the other hand, the results in the same figure show that the introduction of a plasticizer into the pure PLA enhances its moisture resistance. A relatively lower level of moisture absorption is noted for the PLA with the plasticizer.

For the enset–PLA, the absorption reaches a saturation point or plateau after about 26 days; hence, 14 more days are considered after reaching this stable point. In this scenario, the highest water absorption noted is 4.7%; it is found for the highest fiber concentration of 25% and a plasticizer percentage of 6%. This higher absorption is mainly caused by the hydrophilic nature of the fiber in the composite, and this is in agreement with other related findings, though the rate of uptake differs based on the fiber type [4,27]. Since the water absorption rate of PLA is less than 1%, the overall result can be interpreted as a result of the water uptake of the enset or the interface [13,30].

The composite material with the highest fiber ratio shows drastic moisture absorption in the first 5 days, after which the rate begins to decline gradually. Accordingly, more than 20% of the total moisture absorbed by the 25% fiber containing the enset–PLA composite material is absorbed in the first 4 days. The rate starts to decline from 5th day of submerging the specimen in the distilled water until the 25th day. Moreover, stability is reached after 26 days for the same material. Similar trends are noted for other fiber ratios but with a relatively smaller rate and lower absorption.

Correspondingly, the lowest water absorption (1.3%) of the enset-fiber-containing composites is found for the lowest fiber ratio of 15% and the plasticizer ratio of 6%. The moisture absorption decreases with the decreasing fiber concentration and increasing plasticizer ratio according to other related findings [12,40,42].

The moisture release trend and residual moisture are assessed after characterizing the absorption. The release becomes nearly stable after 13 days for all the considered materials. The scenario with the highest moisture release rate happens for the highest absorbing composite, the 25% fiber, which is in agreement with other findings [40,43]. Figure 3 below shows the moisture release trend at room temperature with the residual moisture compared to that of the dried specimen before submerging it into water.

From the result presented in the above figure, about 80% of the moisture absorbed is released in the first 10 days of exposure to room temperature. The material with the higher fiber concentration shows a relatively higher rate of release. When exposing the specimen to room temperature, 0.1% to 0.35% residual moisture is noted even after the stability point at which there is no more release. This stability is monitored for about 10 days to determine the amount of residual moisture.

### 3.2. Effect of Plasticizer Ratio on Moisture Absorption

Figure 2 above indicates that the presence of a plasticizer in pure PLA decreases the moisture abortion and the rate thereof. For the PLA-containing fiber, the effects of the plasticizer on the moisture uptake are significant. In this regard, adding a 6% plasticizer to the pure PLA decreases the moisture uptake by 18.3%. However, going above this percentage of plasticizer deteriorates its mechanical characteristics, though it may be advantageous in terms of moisture resistance. On the other hand, compounding becomes difficult when increasing the plasticizer due to sticking of the pellets.

Considering different scenarios, increasing the plasticizer ratio enhances the moisture resistance. For instance, an increase in the plasticizer from 4% to 6% results in a 0.5% lower moisture uptake for the 25% fiber ratio of the composite. This equals about 11% of the overall moisture absorption. This trend is similar for the other fiber ratios considered, though the amount differs. These results are in agreement with other findings stating that the presence of a plasticizer reduces moisture absorption [40,43]. It should be noted that the literature reports some variation in the total decrease in moisture uptake with respect to the fiber and plasticizer content. The reason behind this might be the properties of the fibers and their concentration thereof in the composite, as well as the processing and testing conditions under consideration.

### 3.3. Effects of Moisture on Mechanical Properties

The subsequent property losses due to the absorbed moisture were assessed. Figures 4 and 5 present the subsequent effect of moisture on the mechanical properties of the enset–PLA composite jointly with a comparison of before and after its submersion in water. After the moisture absorption, the tensile strength and stiffness of the materials showed a decline. The highest deterioration occurred for the high fiber concentration and the lowest plasticizer ratio. In this regard, the maximum property loss happened for the 25% fiber with the 6.25% plasticizer; the tensile strength and stiffness decreased by 16.7% and 20%, respectively. On the other hand, our full results show that the tensile strength decreases due to moisture absorption are 11.2%, 13.9%, and 16.7% for 15%, 20%, and 25% enset fiber ratios of composites containing PLA.

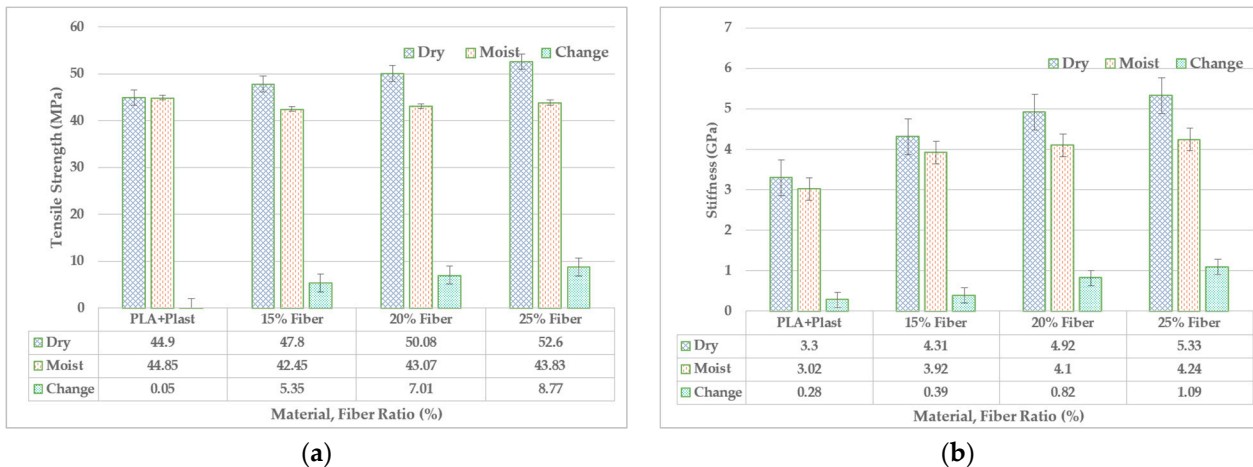

(**a**)  (**b**)

**Figure 4.** Effects of moisture on mechanical properties (**a**) on tensile strength and (**b**) on modulus.

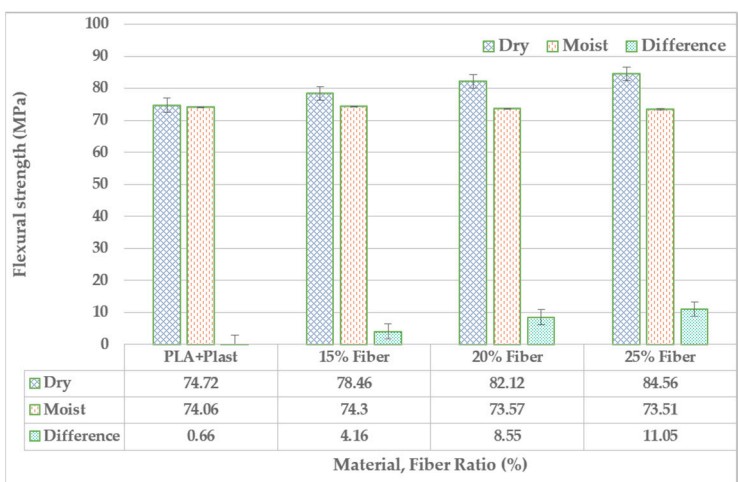

**Figure 5.** Effects of moisture on bending strength of enset–PLA.

The level of the deterioration of the mechanical properties of the enset–PLA composite is slightly better than that of some other PLA-based bio-composites exposed to a similar environment [44]. For instance, an injection-molded PLA-cordenka composite exhibits a smaller tensile strength when compared with an enset–PLA composite under similar conditions. This difference might have emanated from the type and properties of the fibers as well as the processing condition.

The enset–PLA composite with a loss of properties due to moisture has as good properties as those of PLA–cordenka and PLA–flax composites before a similar degradation resulting from moisture. For instance, the PLA–cordenka composite shows strengths of 50.4 Mpa, 50.7 Mpa, and 57.9 MPa without a loss of properties due to moisture for fiber ratios of 10, 20, and 30%, respectively. When compared for the same fiber concentration (20%), the strengths of degraded PLA–enset and PLA–cordenka composites are almost equal (about 51 Mpa). Likewise, the flax–PLA composite shows comparable characteristics to those of the moisture-degraded enset–PLA composite, with values of 42.7 Mpa, 49.2 Mpa, and 54.1 Mpa for the 10%, 20%, and 30% fiber ratios, respectively. This attribute of the enset–PLA composite—its appreciable strength even after a loss in its properties due to moisture—might make it suitable for automotive interior part development for which moisture effects are expected due to the difference between the car's interior and exterior environment. This could be inspiration for further studies considering each part. Additionally, moisture-exposed environments could also be considered. Enset–PLA composites still lose some of their properties due to the effect of moisture. This might be triggered when moisture causes

swelling, de-bonding, bond breaking, and internal stress because of the fiber concentration and its associated hydrophilicity [18,19,24].

However, the stiffness of the enset–PLA composite is affected differently. An increase in the fiber concentration significantly enhances the stiffness both before and after the moisture uptake despite the difference in the percentage of increase. Unlike the tensile strength, increasing the fiber ratio significantly enhances the stiffness in both conditions. As is evident from Figure 4 above, the highest possible stiffness is achieved for the material containing 25% fiber, with greater than 5.3 Gpa and 4.3 Gpa for the dry and moist fiber, respectively.

Similarly, the flexural characteristics of the enset–PLA composite have to be scrutinized. There is a slight difference in the degradation resulting from moisture. Unlike the tensile properties explained above, the loss of flexural properties due to moisture uptake is comparable and is almost equal to the property gain resulting from the increasing fiber concentration before the moisture uptake. Figure 5 shows that the strengths of the 15%, 20%, and 25% fiber concentrations do not show a significant difference after the absorption of moisture. The bending strengths after the degradation emanating from moisture of all five materials plotted in Figure 5 below are very similar. The flexural strength that has been gained because of the increase in the fiber concentration is almost equivalent to the strength lost due to the exposure to moisture.

Though the flexural strength is affected due to moisture and the rate increases with the increasing fiber ratio, the loss in strength is compensated by the gain emanating from the presence of the fiber. For instance, adding 15% fiber to the PLA–plasticizer allows the material to gain about 5.3% more strength, but exposing it to moisture until it reaches a stability point decreases its strength by 5.1%. Again, it takes the addition of 20% enset fiber to increase the bending strength by 13.1% in a dry condition, while exposing it to moisture until a plateau point is reached reduces its gained strength by about 12.8%. The flexural strength decreases due to moisture absorption are 5.3%, 10.4%, and 13.1% for the 15%, 20% and 25% enset fiber ratios of composites containing PLA.

For the enset–PLA material compared in Figure 5 above, the maximum and minimum flexural strength losses for the fiber containing the PLA–plasticizer matrix are 13.1% and 5.3%, and they are found for the 25% and 15% fiber ratio composites, respectively. Increasing the fiber ratio for the composite exposed to moisture does not increase the flexural strength noticeably. This is might have resulted from the effect of hydrophilic fiber in the composite, resulting in interfacial bond weakening and the swelling up of the specimen under flexural loading [42,44].

### 3.4. Statistical Analysis and Result Summary

A statistical analysis was performed to find the effects of individual factors on the moisture absorption. The main and interactive effects of up three levels were checked. While doing this, the significance level of the factors, an analysis of variance, and a desirability check were conducted. The results can be summarized as follows.

- As for the main effect of individual factors, the fiber ratio and plasticizer ratio are found to significantly affect the moisture absorption and release characteristics of enset–PLA composites, though their effects are inversely related. From the above discussion, it is evident that the increase in the plasticizer ratio enhances the moisture resistance while the increase in fiber increases the hydrophilicity of the composite. Hence, the optimum combination of plasticizer and fiber in the PLA also has a subsequent loss of properties. In this study, attractive and comparable results were achieved with the 4% and 6% plasticizers. Based on our preliminary test results mentioned in the methods section, the age of the plant from which the fiber is extracted and the fiber length are less significant.
- Degradation due to moisture decreases the tensile and flexural strength significantly. The gain in properties due to the increase in fiber ratio is lost due to exposure to moisture for both strength values. The tensile strength becomes slightly greater when

compared with the flexural strength gain due to the fiber ratio when exposure to moisture. However, the effect of moisture on the tensile modulus is less.

- The combined effect of the plasticizer ratio, fiber ratio, and residual moisture on the tensile and bending strength is significant. This can also be seen in Figures 4a and 5 above. One of the reasons that the tensile strength and flexural strength show insignificant increases with increasing fiber ratios is related to the effect of the aforementioned factors. Increasing the fiber ratio significantly increases both the tensile and flexural strength before submerging the test specimen into water and allowing it to absorb moisture.

- Therefore, the following summary can be made from the above experimental and statistical results. Plasticizer utilization in PLA is required due to its brittle nature. Increasing the plasticizer ratio up to 6% is optimal; good moisture resistance is achieved and the subsequent loss of properties is also relatively minimal. Decreasing the plasticizer below 4% decreases the moisture resistance while relatively increased strength values are noted. On the other hand, increasing the plasticizer above 6% significantly decreases the strength, though increasing the moisture resistance. Still, as we noticed during the compounding, exceeding 6% makes the compounding difficult for enset–PLA composites due to the sticking of the pellets. This was also a challenge during the drying before the injection molding process to produce the specimens, during which the pellets stuck to each other. Again, the flexural and tensile strength with the stiffness of a material increased with increasing fiber concentrations. According to the results of this study, a 25% fiber ratio results in the greatest strength both in dry and moist conditions. The amount of strength gained as a result of increasing fiber ratio in moist conditions is less than that of dry condition.

## 4. Conclusions

From the all of the aforementioned results, the following conclusions can be drawn. Enset fiber is an attractive natural fiber that can be compounded with PLA to result in competitive and slightly better mechanical properties compared with those of other selected natural fibers. It gives competitive values of tensile and flexural strength before and after exposure to moisture and the loss of some of its mechanical properties. In this regard, the fiber ratio affects the hydrophilic nature of the enset–PLA composite, though the rate of absorption varies based on the fiber type and ratio. Though the bending strength increases with the increasing fiber ratio without submerging it into moisture, it is not increased with the increasing fiber ratio after exposure to moisture. The amount of property gain from adding more fiber is compensated by the amount of property loss resulting from the hydrophilicity introduced by the same added fiber ratio. On the other hand, no significant change in tensile strength is noted for the composites when increasing the fiber ratio for moisture-absorbing test specimens. Unlike the tensile strength, increasing the fiber ratio significantly enhances the stiffness both with and without moisture absorption. The individual effects of the fiber ratio and plasticizer concentration in composites of enset–PLA are significant.

**Author Contributions:** Conceptualization, A.A., F.D. and B.B.; methodology, A.A., F.D., B.B., M.V. and S.H.; Software, A.A.; Validation, A.A., F.D., B.B. and B.S.; formal analysis, A.A., F.D., B.B. and B.S.; formal analysis, A.A., F.D. and B.B.; investigation, A.A., F.D., B.B. and B.S.; resources, A.A., F.D., B.B., M.V. and S.H.; data curation, A.A.; writing—A.A.; writing—review and editing, F.D., B.B., B.S. and A.A.; visualization, A.A.; supervision, F.D., B.B. and B.S. All authors have read and agreed to the published version of the manuscript.

**Funding:** This research received no external funding.

**Institutional Review Board Statement:** Not applicable.

**Informed Consent Statement:** Not applicable.

**Data Availability Statement:** Not applicable.

**Acknowledgments:** We are very grateful to Mekelle University (Ethiopian Institute of Technology-Mekelle), KU Leuven, and Addis Ababa Science and Technology University (AASTU) for granting us the privilege of using their lab facilities for different tests. We are also thankful to all the lab technicians in these institutions together with the students working therein for their unreserved cooperation and help.

**Conflicts of Interest:** The authors declare no conflict of interest.

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
