# Peer review of "Moisture Absorption Characteristics and Subsequent Mechanical Property Loss of Enset–PLA Composites"

_jcs, doi:10.3390/jcs7090382_

Round 1

Reviewer 1 Report

The researchers examine moisture's effect on Enset-PLA composite mechanics. They craft varied specimens with different factors, track moisture absorption, perform mechanical tests, and analyze. Results show that despite moisture, composites maintain stability and competitive properties. For example, a 11% decrease in tensile strength and 5% in bending strength occurs for a 15% fiber volume fraction due to moisture. However, it is crucial to make minor changes and respond to inquiries before contemplating publication.

1-      How do the authors vary the composite specimens in terms of fiber volume fraction (e.g., 25%, 10%), plasticizer ratio, fiber length, and fiber age?

2-      Can you describe the procedure employed to subject the composite specimens to moisture absorption? Why is it important to monitor moisture uptake on a daily basis?

3-      Which mechanical properties, such as tensile strength, bending strength, and strain at break, are investigated to assess the composite's behavior under different conditions?

4-      How does the authors' utilization of an optimal design of experiment aid in understanding the distinct and combined effects of factors like fiber volume fraction, plasticizer ratio, fiber length, and fiber age?

5-      What are the specific findings regarding the influence of moisture on the mechanical properties of the Enset-PLA composites? For instance, what percentage decrease in tensile strength and bending strength is observed for a 15% fiber volume fraction due to moisture?

6-      How does the plasticizer ratio impact the composites' resistance to moisture absorption, and is there a quantifiable relationship established in the study?

7-      Based on the results, how do the authors conclude that Enset-PLA composites demonstrate competitive properties and stability, even in the presence of moisture?

8-      In practical applications, how might these research insights guide the selection and use of Enset-PLA composites when moisture exposure is a consideration?

The English language in your text displays a commendable level of quality. Your sentences are skillfully constructed, successfully conveying the intended message.

Author Response

Dear Sir,

Thank you for your valuable comments.

I have attached the point by point responses to that comments herewith.

King Regards,

Abebayehu

Reviewer 2 Report

This manuscript by Abdela et. al. presented a study that considers the PLA-Enset composite and typifies hygroscopic characteristics and their resulting mechanical properties loss. the whole manuscript was well-organized, and the information provided in this study and the experimental methodology are interesting. However, the authors could have explained this manuscript more thoroughly. Overall, I think this manuscript is suitable for publication in JCS after major revision.

Ensure that the introduction clearly establishes the importance of studying moisture absorption in Enset-PLA composites and its impact on mechanical properties.

Clarify the rationale for selecting specific parameters such as fiber volume fraction, plasticizer ratio, fiber length, and fiber age. Explain how these factors relate to real-world scenarios.

Explain the process of the optimal design of experiments more comprehensively. Describe how the factors were varied and what specific insights were gained from the analysis.

Summarize the key findings succinctly and emphasize the practical implications for using Enset-PLA composites in real-world conditions

Author Response

Dear Sir,

Thank you very much for your valuable comments.

I have attached the point by point review responses and correction herewith.

Kind Regards,

Abebayehu 

Round 2

Reviewer 1 Report

I'd like to express my gratitude for the clarification you provided regarding these aspects. Upon examining the changes you pointed out in the latest version of the paper, I can confirm that the authors' revisions are articulated clearly, follow a logical progression, and are quite convincing. I haven't come across any errors in these particular areas.

The English language in your text displays a commendable level of quality. Your sentences are skillfully constructed, successfully conveying the intended message.

Reviewer 2 Report

The authors have addressed in detail the concerns raised by the referees and the manuscript is suitable for publication.